# Phase-Controlled Tunable Unconventional Photon Blockade in a Single-Atom-Cavity System

**DOI:** 10.3390/mi14112123

**Published:** 2023-11-19

**Authors:** Hong Li, Ming Liu, Feng Yang, Siqi Zhang, Shengping Ruan

**Affiliations:** 1State Key Laboratory on Integrated Optoelectronics and College of Electronic Science and Engineering, Jilin University, Changchun 130012, China; quantum208@163.com; 2Institute for Interdisciplinary Quantum Information Technology, Jilin Engineering Normal University, Changchun 130052, China; 18943152956@163.com (M.L.); yangfeng@jlenu.edu.cn (F.Y.); siqizhang88@163.com (S.Z.)

**Keywords:** phase, photon blockade, second-order correlation function, single-atom-cavity system

## Abstract

In the past few years, cavity optomechanical systems have received extensive attention and research and have achieved rapid development both theoretically and experimentally. The systems play an important role in many fields, such as quantum information processing, optomechanical storage, high-precision measurement, macroscopic entanglement, ultrasensitive sensors and so on. Photon manipulation has always been one of the key tasks in quantum information science and technology. Photon blockade is an important way to realize single photon sources and plays an important role in the field of quantum information. Due to the nonlinear coupling of the optical force system, the energy level is not harmonic, resulting in a photon blockade effect. In this paper, we study the phase-controlled tunable unconventional photon blockade in a single-atom-cavity system, and the second-order nonlinear crystals are attached to the cavity. The cavity interacts with squeezed light, which results in a nonlinear process. The system is driven by a complex pulsed laser, and the strength of the coherent driving contains the phase. We want to study the effect of squeezed light and phase. We use the second-order correlation function to numerically and theoretically analyze the photon blockade effect. We show that quantum interference of two-photon excitation between three different transition pathways can cause a photon blockade effect. When there is no squeezed light, the interference pathways becomes two, but there are still photon blockade effects. We explore the influence of the tunable phase and second-order nonlinear strength on the photon blockade effect. We calculate the correlation function and compare the numerical results with the analytical results under certain parameters and find that the agreement is better.

## 1. Introduction

In cavity optomechanics and its related fields, people always try to control the quantum-mechanical interactions between electromagnetic radiation and macroscopic mechanical resonance. Soviet physicist Braginsky studied the effect of light on mechanical oscillators [1]. Subsequently, some researchers have proposed that this system can be used to achieve precise measurement of the weak force [2]. In 1980, Dorsel in Germany suspended one end of the cavity mirror and successfully observed the forced vibration of the suspended mirror as a mechanical oscillator under light pressure in the cavity [3]. Cavity optomechanics has developed rapidly theoretically and experimentally [4,5,6]. The ground state cooling of a mechanical oscillator can be realized by using an optical force cavity [7,8]. The optical mode can be modulated by mechanical mode, for example, an optical force cavity is used to achieve optical force entanglement [9,10], photon blockade [11] and optical force-induced transparency [12,13]. This provides a great platform for quantum information and quantum control in recent years. Typically, in optomechanical systems, there are optical and mechanical modes, and they interact with each other, and these modes can be in the frequency range of terahertz and megahertz bands, respectively. As such, it provides a good bridge between quantum information science and optical communication.

As the most important resource, a single-photon source plays a crucial role in quantum communication and quantum computing [14,15]. Therefore, how to realize the single-photon source has been the focus of attention, and the photon blockade (PB) effect is one of the most effective methods [16,17,18]. The concept of photon blockade was first proposed by Imamoğlu in 1997 [19], and the basic idea of photon blockade is similar to electron coulomb blockade in physics [20]. In 2005, the Kimble group of the United States experimentally verified the photon blockade effect in the cavity atomic system for the first time [21]. The experiment measured the intensity correlation function and found that photons exhibit sub-Poisson statistical properties and an anti-bunching effect. There are two types of photon blockade mechanisms: conventional photon blockade (CPB) and unconventional photon blockade (UPB). Since its conception, the photon blockade effect has been observed in different experimental systems. The CPB mechanism generates a single-photon source because of the presence of anharmonicity in energy levels of the system. This requires a large nonlinear coupling in order to produce an inharmonic in the energy levels. In quantum physics, the generation and manipulation of nonclassical light has become a hot topic [22,23,24,25]. In recent years, photon blockade technology has attracted great interest, and this typical nonclassical light can produce anti-bunching photons. Single-photon blockade can be understood as the presence of a single photon preventing the generation of a second photon in a nonlinear cavity driven by a classical light field [26,27,28]. Due to the fact that a single-photon source is widely used in quantum communication and the quantum information field, it has been widely studied in recent years [29,30,31,32,33,34,35,36,37,38]. For example, CPB has been predicted in a coupled cavity quantum system [39,40,41,42,43], quantum gates [44] and a second-order nonlinear cavity system [45,46,47,48,49].

In 2011, Bamba and Ciuti of France found out through theoretical analysis that the physical mechanism of this kind of photon blockade effect is the destructive quantum interference between different transition paths of photons in the coupled Kerr cavity system [50]. In order to distinguish between how these two different physical mechanisms produced photon blockade, in 2013, Carusotto of Italy named the photon blockade caused by destructive quantum interference between different transition paths as unconventional photon blockade [51]. UPB breaks the limit of CPB to nonlinear strength, reveals the weak nonlinear and can realize the photon blockade effect. The UPB effect has been predicted for the first time in the experiment from Ref. [52], showing a quantum dot cavity QED system. In the two weakly nonlinear coupled cavities, quantum interference between the different paths prevents the presence of two photons [53,54,55]. This effect has recently been considered in a variety of models, such as coupled cavities with Kerr-type nonlinearity [56,57,58,59,60,61], optical cavity with a quantum dot [62] and so on.

Previous studies on photomechanics mainly focus on the UPB without the phase, but phase is an extremely important parameter for unconventional photon blockade and even basic quantum mechanics [63,64,65]. Motivated by this, we intend to study the phase-controlled tunable UPB in a single-atom-cavity system. In this paper, we numerically and theoretically analyze the photon blockade effect in a coupled single-atom-cavity system, which includes a tunable phase of complex drive strength and second-order nonlinear crystals. We explore the influence of the tunable phase and second-order nonlinear strength on photon blockade effect. We have studied the second-order correlation function and compare the numerical results with the analytical results under certain parameters and find that the agreement is better.

The paper is arranged as follows. In Section 2, we introduce the physical model and calculate the dynamical equations of the system. In Section 3, we illustrate the equal-time-second-order correlation function by solving the master equation numerically and analyze the photon blockade characteristics. We then compare the analytical results with the numerical results. Finally, in Section 4, we give the conclusion.

## 2. Model and Dynamical Equations of the System

As shown in Figure 1a, we took into account a nonlinear optomechanical system that contains a cavity mode with frequencies $\omega_{c}$ and a two-level system with frequencies $\omega_{a}$. The system is driven by a complex pulsed laser, the cavity exhibits second-order nonlinear crystals and the strength is *U*. In a rotating frame, the transformed Hamiltonian is obtained as [66]
(1)$$H=\Delta_{c}a^{†}a+\Delta_{a}\sigma^{†}\sigma +g\left(\sigma a^{†}+\sigma^{†}a\right)+\Omega \left(a^{†}e^{i\varphi }+ae^{-i\varphi }\right)+\frac{U}{2}\left(a^{†2}+a^{2}\right),$$
where $a(a^{†})$ is the annihilation (creation) operator for the optical mode of the cavity and $\sigma (\sigma^{†})$ is the lowering (raising) operator of the two-level atom system. $\Omega e^{i\varphi }$ is a tunable phase of complex driving strength, Ω is the amplitude of the driving field, φ is the phase of the driving field and *g* is the strength of the interaction between photons and atoms. $U=\chi^{2}U_{p}$, where $\chi^{2}$ is the second-order nonlinear crystals and $\chi^{2}$ is driven by the amplitude field $U_{p}$. We introduce the detunings of the cavity and atom $\Delta_{c}=\omega_{c}-\omega_{l}$ and $\Delta_{a}=\omega_{a}-\omega_{l}$, where $\omega_{l}$ is the driving pump frequency.

The statistical properties of photons are described by the second-order correlation function in steady state, which is given by
(2)$$g_{a}^{\left(2\right)}\left(0\right)=\frac{\langle a^{†}a^{†}aa\rangle }{{\langle a^{†}a\rangle }^{2}}.$$

In general, the value of the second-order correlation function $g_{a}^{\left(2\right)}\left(0\right)$ is calculated to determine whether the photon anti-bunching effect occurs. When $g_{a}^{\left(2\right)}\left(0\right)>1$, the photon arrival detection field is in groups and we can say that the light field appears to have bunching effect, showing a super-Poisson distribution, which greatly increases the probability of two-photon existence in the cavity. Conversely, $g_{a}^{\left(2\right)}\left(0\right)<1$ indicates that the anti-bunching effect of the light field shows sub-Poisson distribution and a photon blockade effect occurs, which effectively inhibits the probability of two-photon existence in the cavity. If $g_{a}^{\left(2\right)}\left(0\right)\to 0$, it means that the system is in a complete photon blockade mechanism, and the probability of two photons in the cavity at the same time is close to zero.

Considering the dissipation of the system, the system dynamic evolution process can be described by the Lindblad master equation, i.e.,
(3)$$\dot{\rho }\left(t\right)=i\left[\rho ,H\right]+\sum_{k=a,b}L_{k}\left(\rho \right).$$

The first term in the equation is the Schrödinger equation term, which represents the unitary evolution, and the other terms represent the system’s dissipation, transition and decoherence, which are caused by the interaction between the system and the environment. The Lindblad super operators are defined by $L_{a}\left[\rho \right]=\frac{\kappa }{2}\left(2a\rho a^{†}-a^{†}a\rho -\rho a^{†}a\right)$ and $L_{b}\left[\rho \right]=\frac{\gamma }{2}\left(2\sigma \rho \sigma^{†}-\delta \sigma^{†}\sigma \rho -\rho \sigma^{†}\sigma \right)$, where κ and γ are the decay rates of the cavity and the two-level atom, respectively. The Hamiltonian *H* is given in Equation (Equation 1). The steady-state solutions of the master equation are obtained by numerical solutions,
(4)$$g_{a}^{\left(2\right)}\left(0\right)=Tr\left(\rho a^{†}a^{†}aa\right)/{\left[Tr\left(\rho a^{†}a\right)\right]}^{2}.$$

When the driving field is very weak, Ω/κ≪1, the wave function of the system can be expanded as
(5)$$|\psi \rangle ≃C_{g0}|g0\rangle +C_{g1}|g1\rangle +C_{g2}|g2\rangle +C_{e0}|e0\rangle +C_{e1}|e1\rangle ,$$
where $C_{mn}\left(t\right)$ for m=g,e and n=0,1,2 represents the probability amplitudes of the bare state |mn〉, the first position in the ket notation $|mn\rangle$ corresponds to the atomic state and the second to the photonic state. The effective non-Hermitian Hamiltonian containing the optical decay κ and the two-level atom with the decay rates γ is
(6)$$\begin{array}{ccc} H^{\prime } & = & \left(\Delta_{c}-i\frac{\kappa }{2}\right)a^{†}a+\left(\Delta_{a}-i\frac{\gamma }{2}\right)\sigma^{†}\sigma +g\left(\sigma a^{†}+\sigma^{†}a\right)\\ & & +\Omega \left(a^{†}e^{i\varphi }+ae^{-i\varphi }\right)+\frac{U}{2}\left(a^{†2}+a^{2}\right). \end{array}$$

For a weak drive, based on the effective non-Hermitian Hamiltonian in Equation (Equation 6) and the wave function in Equation (Equation 5), by using Schrödinger equation $i{𝜕}_{t}|\psi \rangle =H^{\prime }|\psi \rangle$, we can obtain series dynamical equations of the coefficients:(7)$$i\frac{𝜕C_{g1}}{𝜕t}=\Omega e^{i\varphi }+gC_{e0}+\left(\Delta_{c}-i\kappa /2\right)C_{g1}+\sqrt{2}\Omega e^{-i\varphi }C_{g2},$$
(8)$$i\frac{𝜕C_{g2}}{𝜕t}=\sqrt{2}gC_{e1}+2\left(\Delta_{c}-i\kappa /2\right)C_{g2}+\sqrt{2}\Omega e^{i\varphi }C_{g1}+\frac{\sqrt{2}}{2}U,$$
(9)$$i\frac{𝜕C_{e0}}{𝜕t}=gC_{g1}+\left(\Delta_{a}-i\gamma /2\right)C_{e0}+\Omega e^{-i\varphi }C_{e1},$$
(10)$$i\frac{𝜕C_{e1}}{𝜕t}=\sqrt{2}gC_{g2}+\left(\Delta_{c}+\Delta_{a}-i\kappa /2-i\gamma /2\right)C_{e1}+\Omega e^{i\varphi }C_{e0}.$$

Under weak driving condition $|C_{g0}\left|\gg \right|C_{g1}\left|,\right|C_{e0}\left|\gg \right|C_{g2}\left|,\right|C_{e1}|$, in the steady state $𝜕C_{mn}/𝜕t=0$, the probability amplitudes can be obtained as
(11)$$\begin{array}{ccc} C_{g1}M & = & 2\Omega [2(p+d)-U(f\cos\varphi +2h\sin\varphi )]\\ & & +i[2\Omega (x-\gamma y)-U(2h\cos\varphi -f\sin\varphi )], \end{array}$$
(12)$$\begin{array}{ccc} C_{g_{2}}M & = & \sqrt{2}\left[U\left(p+d+4\Delta_{c}g^{2}\right)-2\Omega^{2}\left(f\cos\varphi -2h\sin\varphi \right)\right]\\ & & -\frac{i}{\sqrt{2}}\left[U\left(4\gamma g^{2}-x+\gamma y\right)+4\Omega^{2}\left(2h\cos\varphi +f\sin\varphi \right)\right]. \end{array}$$

The *M* is defined as
(13)M=α+iβ,
where
(14)$$\begin{array}{ccc} \alpha & = & 16g^{4}+\kappa \gamma^{3}+4\left(\Delta_{c}^{2}-\Omega^{2}\right)\left[4\Delta_{a}\left(\Delta_{a}+\Delta_{c}\right)-4\Omega^{2}-\kappa^{2}\right]+4\gamma \kappa \left[\Omega^{2}-\Delta_{c}\left(3\Delta_{c}+4\Delta_{a}\right)\right]\\ & & +\gamma^{2}\left[4\Omega^{2}+\kappa^{2}-4\Delta_{a}\left(3\Delta_{c}+\Delta_{a}\right)\right]+4g^{2}\left[\gamma^{2}+2\gamma \kappa -4\left(\Delta_{c}^{2}+2\Delta_{c}\Delta_{a}+2\Omega^{2}\right)\right], \end{array}$$
(15)$$\begin{array}{ccc} \beta & = & 16g^{2}\left[\gamma \left(\Delta_{a}+\Delta_{c}\right)+\kappa \Delta_{c}\right]+2\kappa \gamma^{2}\left(3\Delta_{c}+2\Delta_{a}\right)+4\gamma \Delta_{c}\left[\kappa^{2}-2\Delta_{a}\left(3\Delta_{c}+2\Delta_{a}\right)\right]\\ & & +2\Delta_{a}\gamma^{3}+8\left[\gamma \Omega^{2}\left(2\Delta_{c}+\Delta_{a}\right)-\kappa \left(\Delta_{c}^{2}-\Omega^{2}\right)\left(\Delta_{c}+2\Delta_{a}\right)\right]. \end{array}$$

These coefficients are taken as $d=\Delta_{c}\left(4\Omega^{2}+\kappa^{2}-4\Delta_{a}\Delta_{c}-4\Delta_{a}^{2}\right)$, $f=4\left(\Omega^{2}-g^{2}\right)-4\Delta_{a}\left(\Delta_{a}+\Delta_{c}\right)+\kappa \left(\kappa +\gamma \right)$, $h=\Delta_{a}\gamma +\kappa \left(\Delta_{c}+2\Delta_{a}\right)$, $p=\Delta_{a}\left(4g^{2}+\gamma^{2}\right)+2\gamma \kappa \left(\Delta_{a}+\Delta_{c}\right)$, $x=4\kappa \Delta_{c}\left(\Delta_{c}+2\Delta_{a}\right)-\kappa \left(4g^{2}+\gamma^{2}\right)$ and $y=4\Omega^{2}+\kappa^{2}-8\Delta_{a}\Delta_{c}-4\Delta_{a}{}^{2}$. According to Equations (2) and (5), we approximate the calculation and obtain
(16)$$\begin{array}{c} g_{a}^{\left(2\right)}\left(0\right)=\frac{2P_{g2}}{{\left(P_{g1}+P_{e1}+2P_{g2}\right)}^{2}}≃\frac{2P_{g2}}{P_{g1}^{2}}, \end{array}$$
where $P_{mn}={\left|C_{mn}\right|}^{2}$ and the second-order correlation function $g_{a}^{\left(2\right)}\left(0\right)\ll 1$ means that the system operates in photon blockade effect.

## 3. Numerical Simulation and Results

In this section, the second-order correlation function is numerically studied and the quantum master equation is solved in a truncated Fock space. This article considers the UPB, and the physical mechanism is quantum interference of light between different transition paths. The energy level structure and transition pathways of the system are shown in Figure 1b, and we can see that there are three transition pathways from |g0〉 to |g2〉: the first transition pathway is |g0〉→|g1〉→|g2〉 and the second transition pathway is |g0〉→|g1〉→|e0〉→|e1〉→|g2〉. The existence of nonlinear crystals can lead to a third pathway, |g0〉→|g2〉, driven by the parameter *U*. The quantum interference of light between the three pathways leads to the single photon blockade phenomenon. In order to study the photon statistics in the research system, this section will solve the quantum master equation numerically. To further deepen understanding, we compare the numerical solution with the analytical solution. For convenience, we readjust all parameters according to the dissipation rate of the cavity κ.

In Figure 2, for simplicity’s sake, we kept the total resonance $\Delta_{c}=\Delta_{a}=0$. We plot the $g_{a}^{\left(2\right)}\left(0\right)$ as a function of the normalized coupling strength g/κ for three values of the phase φ. For the system parameters, we choose Ω/κ=0.01, γ/κ=0.5. In Figure 2a, we choose the squeezed light amplitude U/κ=0, and we find that correlation function $g_{a}^{\left(2\right)}\left(0\right)$ curves are the same shape for different values of the phase φ. That is to say, the phase of complex driving strength does not affect the numerical simulation of the second-order correlation function when the U/κ=0. It can be seen from Equations (11)–(15) that *M* does not contain φ, and $C_{g1}$ also does not contain φ when U/κ=0. Although $C_{g2}$ contains φ when U/κ=0, the coefficient is $\Omega^{2}$, and due to the driving field being very weak, the values of trigonometric terms with φ are very small, and the φ values have almost no effect on the correlation function when U/κ=0. In Figure 2b, we choose the squeezed light amplitude $U/\kappa =7\times 10^{-5}$, we find that the three curves are clearly separated and $g_{a}^{\left(2\right)}\left(0\right)$ displays a strong anti-bunching effect at φ=0.8rad. From Figure 2b, we can see that in the presence of nonlinearity, the choice of the field phase considerably affects the correlation function. Therefore, the photon blockade effect can be realized by adjusting the phase φ.

In Figure 3, we show $g_{a}^{\left(2\right)}\left(0\right)$ as a function of the coupling strength $\Delta_{a}/\kappa$ with three different values of the phase φ. For the system parameters, we choose $\Delta_{c}=0$, Ω/κ=0.01, γ/κ=0.5. In Figure 3a, we choose the squeezed light amplitude U/κ=0, and we find that correlation function $g_{a}^{\left(2\right)}\left(0\right)$ curves are the same shape for different values of the phase φ. The reason is the same as Figure 2a. In Figure 3b, we choose the squeezed light amplitude $U/\kappa =7\times 10^{-5}$, we find that the three curves are clearly separated and $g_{a}^{\left(2\right)}\left(0\right)$ displays a strong anti-bunching effect at φ=1.2rad.

In order to more intuitively demonstrate the photon blockade effect, the contour plots of **$\log_{10}g_{a}^{\left(2\right)}\left(0\right)$** as functions of phase φ and detuning $\Delta_{c}/\kappa \left(\Delta_{a}/\kappa \right)$ are shown in Figure 4. In Figure 4a, the parameters are selected as $\Delta_{c}=0$, g/κ=0.5, $U/\kappa =5\times 10^{-5}$ and γ/κ=0.5. In Figure 4b, the parameters are selected as $\Delta_{a}=0$. Other parameters are consistent with Figure 4a. Through the color, we can clearly see which values of $\Delta_{c}\left(\Delta_{a}\right)$ and φ can achieve the strong anti-bunching effect. Similarly, in Figure 5, we plot $\log_{10}g_{a}^{\left(2\right)}\left(0\right)$ as function of phase φ and interaction strength g/κ. In Figure 5a, the system parameters are considered as $\Delta_{c}=\Delta_{a}=0$, Ω/κ=0.01 and U/κ=0. It is clearly seen that the $g_{a}^{\left(2\right)}\left(0\right)$ displays a strong anti-bunching effect at g/κ≈0.43. The value of φ does not affect the result of $g_{a}^{\left(2\right)}\left(0\right)$. In Figure 5b, the values of parameters are considered as $U/\kappa =5\times 10^{-5}$. Other parameters are consistent with Figure 5a. We can see that the $g_{a}^{\left(2\right)}\left(0\right)$ shows a strong anti-bunching effect at g/κ≈0.52 and φ≈0.8rad.

In Figure 6, we discuss the comparison between numerical and analytical solutions of the second-order correlation function. $g_{a}^{\left(2\right)}\left(0\right)$ is plotted as a function of coupling strength g/κ for different *U* and φ. In Figure 6a,b, we compare φ=0 with different values of U/κ, U/κ=0 in Figure 6a and $U/\kappa =10^{-5}$ in Figure 6b. In Figure 6b,c, we compare $U/\kappa =10^{-5}$ with a different values of φ: φ=0 in Figure 6b and φ=0.8rad in Figure 6c. Therefore, the analytic results of the second-order correlation function are almost in agreement with the numerical results obtained by solving the master equation. It is further verified that our results are reliable. As can be seen from Figure 6, there are some minor differences between the analytic results and the numerical results. We think some of the differences may be due to program accuracy and different assumptions in the model description.

## 4. Conclusions

In the introduction, we introduce the background of cavity optomechanical systems and photon blockade. Although strong optical force coupling can produce a strong photon blockade effect, experimentally, it is difficult to achieve strong single-photon optical force coupling in cavity optomechanical systems, and then we add a two-level atom in the cavity to achieve photon blockade effect. The system includes a tunable phase of complex drive strength and second-order nonlinear crystals attached to the cavity. The Hamiltonian of the system consists of the free Hamiltonian, the interacting Hamiltonian, the driving Hamiltonian and the nonlinearity Hamiltonian. There are two transition pathways from |g0〉 to |g2〉 when U/κ=0, but there are three transition pathways |g0〉 to |g2〉 when U/κ≠0. The interference of the different pathways leads to the single photon blockade phenomenon. In order to study the photon statistics in the research system, we solve the Schrödinger equation and the quantum master equation for the correlation function and we study the dynamical equation of the system. It was found that when the amplitude of the squeezed light U/κ=0, the value of φ has no effect on the second-order correlation function. This is also verified by the analytical solution. The value of φ has a great influence on the second-order correlation function when U/κ≠0. The photon blockade effect can be improved by close assessment of the system parameters, as shown in this paper. By analytical and numerical calculations, we show that the second-order correlation function depends on the tunable phase of complex driving strength, and the phase φ and the squeezed light strength *U* can also be used to control the photon blockade. As a result, an extremely strong anti-bunching effect can be achieved, resulting in a value of the second-order correlation function smaller than unity. This study has important guiding significance for the design of micromechanical devices such as single-photon source devices.

## Figures and Tables

**Figure 1 micromachines-14-02123-f001:**
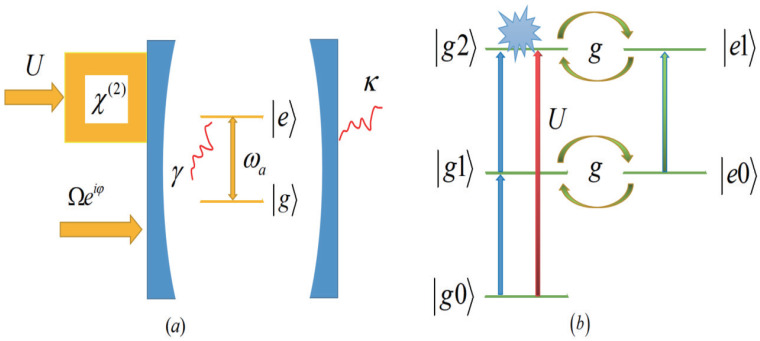
(**a**) Sketch for PB of a tunable phase drive and $\chi^{2}$-type nonlinear crystals in the system. (**b**) Energy-level diagram of the system and the transition pathways for different photon states.

**Figure 2 micromachines-14-02123-f002:**
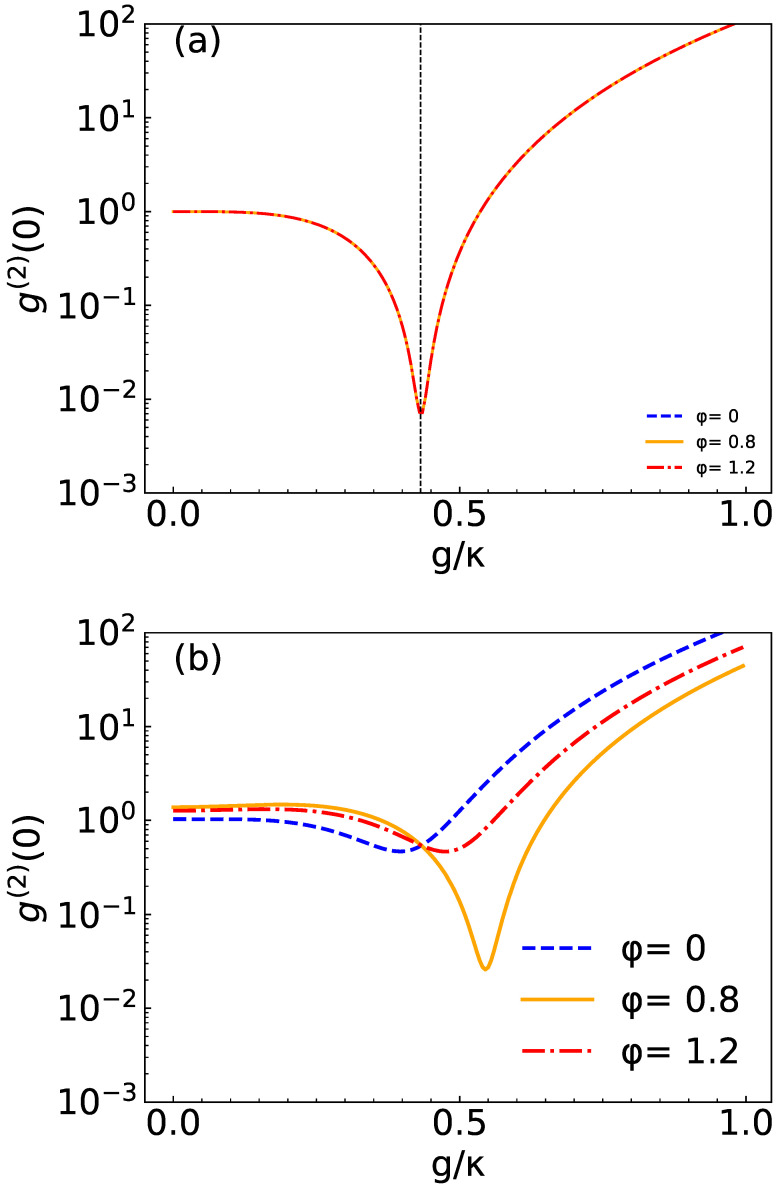
Plot of $g_{a}^{\left(2\right)}\left(0\right)$ at total resonance $\Delta_{c}=\Delta_{a}=0$ as a function of normalized coupling strength g/κ for different values of φ=(0,0.8,1.2)rad. (**a**) U/κ=0 and (**b**) $U/\kappa =7\times 10^{-5}$. Here, we choose Ω/κ=0.01 and γ/κ=0.5, respectively.

**Figure 3 micromachines-14-02123-f003:**
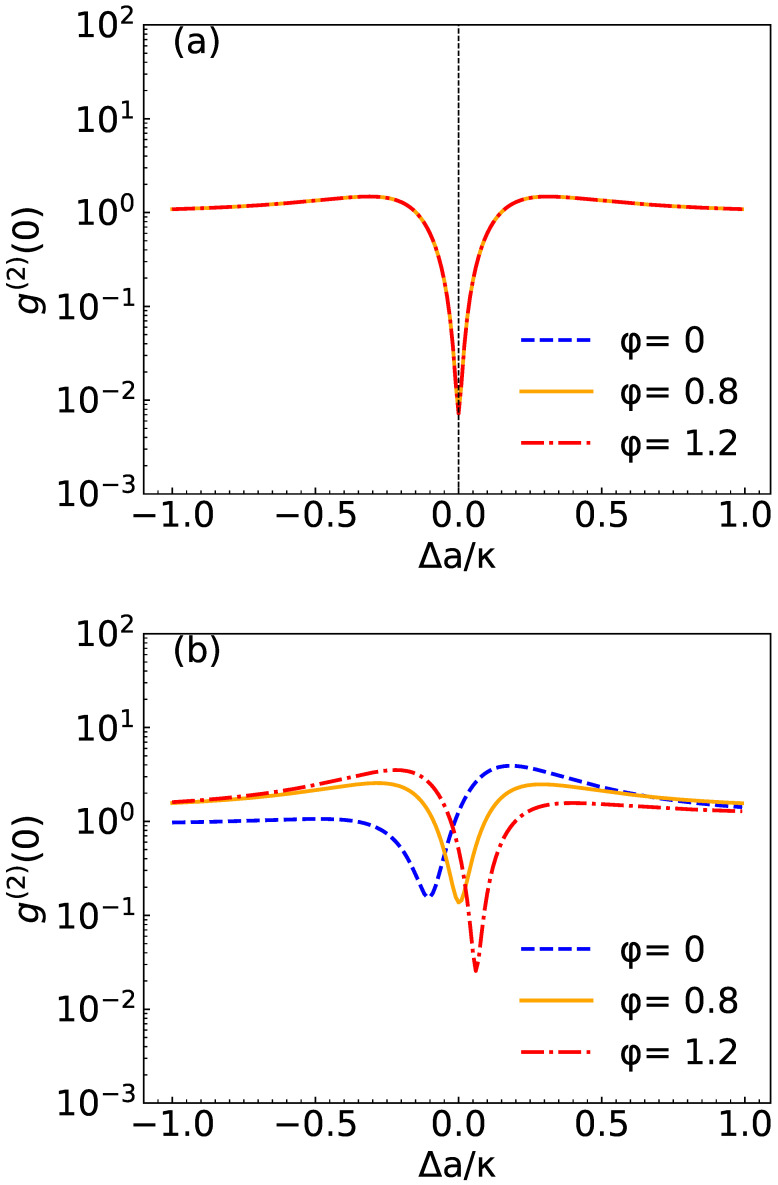
Plot of $g_{a}^{\left(2\right)}\left(0\right)$ at $\Delta_{c}=0$ as a function of detuning $\Delta_{a}/\kappa$ for different values of φ=(0,0.8,1.2)rad. (**a**) U/κ=0 and (**b**) $U/\kappa =7\times 10^{-5}$. System parameters are g/κ=0.43, Ω/κ=0.01 and γ/κ=0.5.

**Figure 4 micromachines-14-02123-f004:**
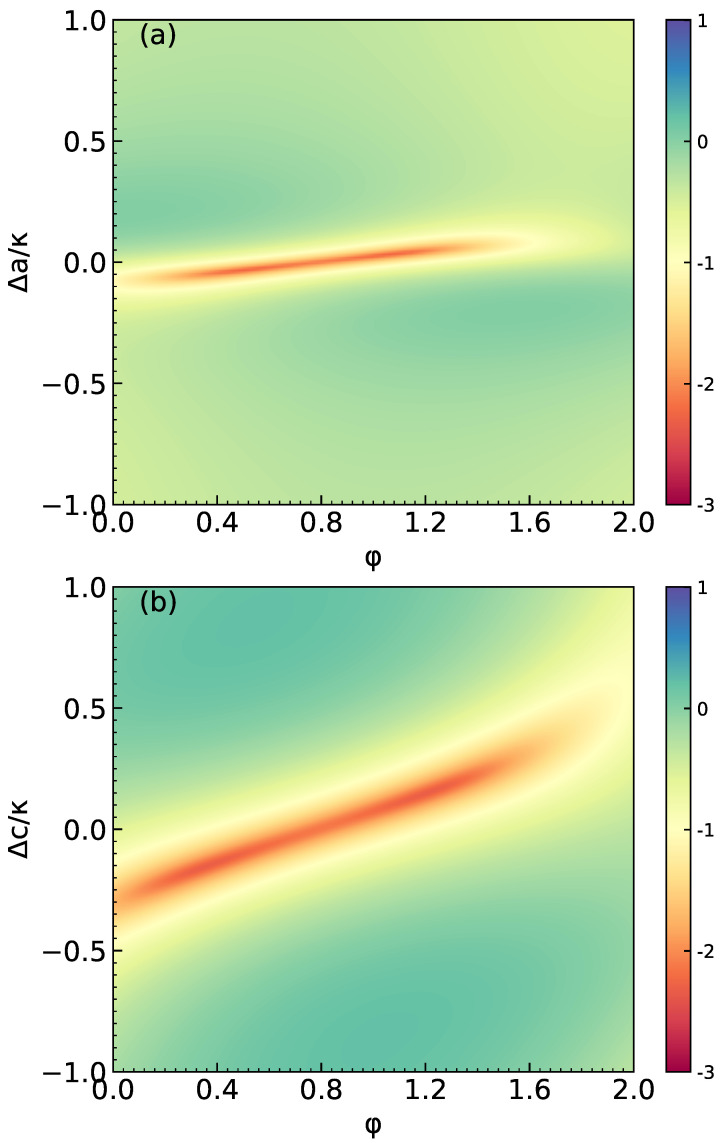
(**a**) Contour plots of $\log_{10}g_{a}^{\left(2\right)}\left(0\right)$ versus φ and $\Delta_{a}/\kappa$ for $\Delta_{c}=0$. (**b**) Contour plots of $\log_{10}g_{a}^{\left(2\right)}\left(0\right)$ versus φ and $\Delta_{c}/\kappa$ for $\Delta_{a}=0$. System parameters are g/κ=0.5, $U/\kappa =5\times 10^{-5}$ and γ=κ/2.

**Figure 5 micromachines-14-02123-f005:**
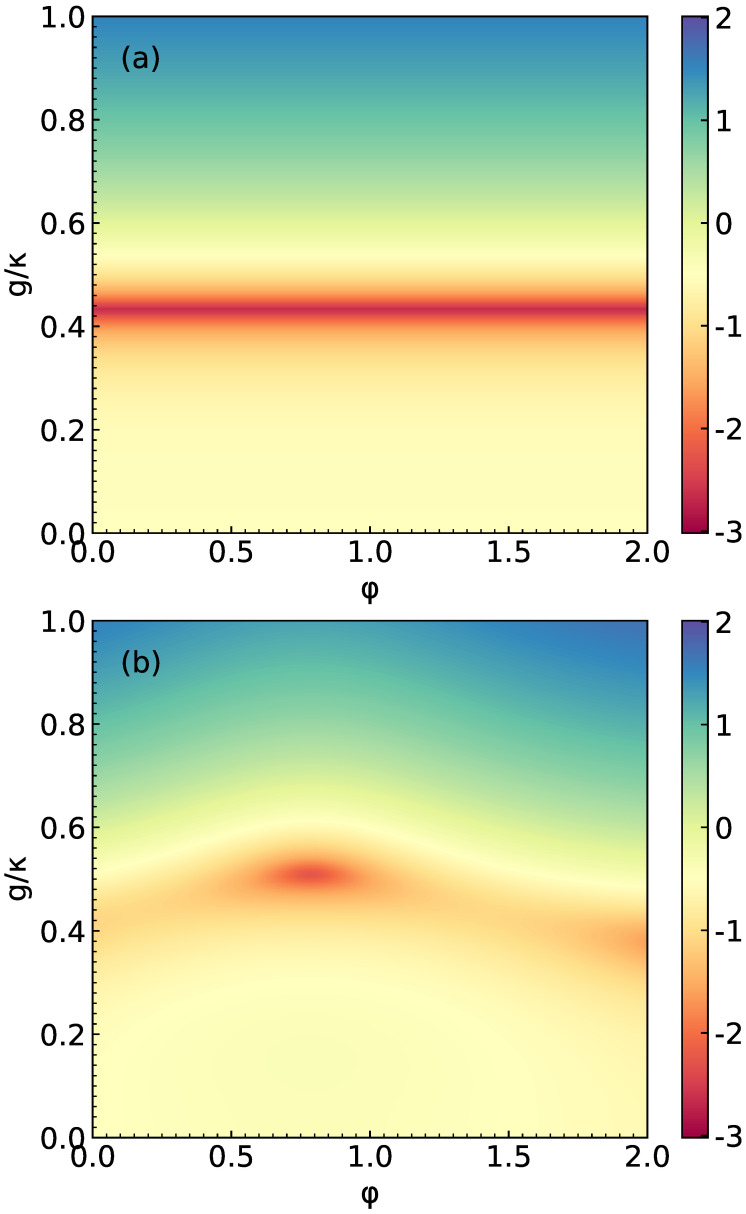
Contour plots of $\log_{10}g_{a}^{\left(2\right)}\left(0\right)$ versus φ and g/κ for different values of U/κ, considered as U=0 in (**a**) and $U/\kappa =5\times 10^{-5}$ in (**b**). Here, we choose $\Delta_{c}=\Delta_{a}=0$, Ω/κ=0.01 and γ=κ/2.

**Figure 6 micromachines-14-02123-f006:**
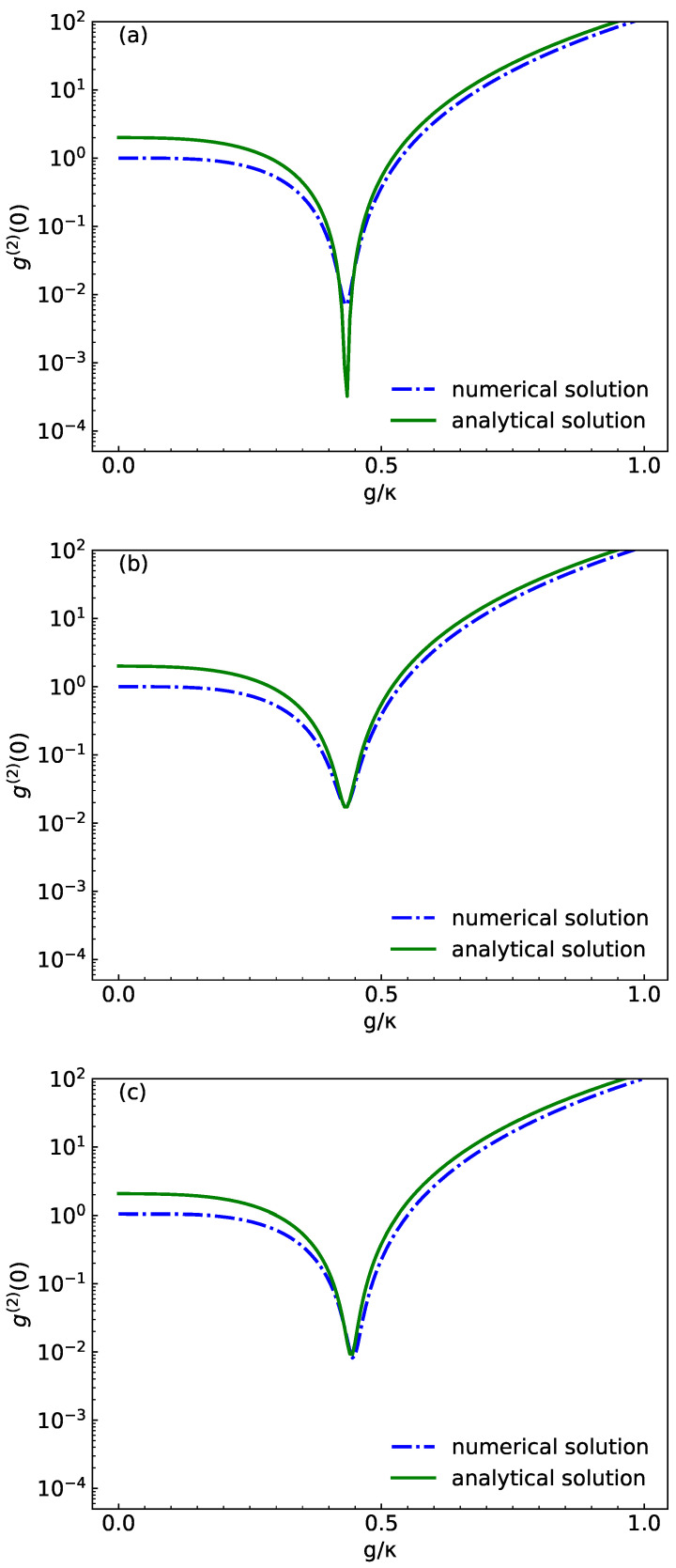
Numerical (blue dots) and analytical (green line) results for $g_{a}^{\left(2\right)}\left(0\right)$ as a function of normalized coupling strength g/κ. The phase and the squeezed light strength are U/κ=0, φ=0 in (**a**), $U/\kappa =10^{-5}$, φ=0 in (**b**) and $U/\kappa =10^{-5}$, φ=0.8rad in (**c**). Here, we choose $\Delta_{c}=\Delta_{a}=0$, Ω/κ=0.01 and γ/κ=0.5.

## Data Availability

Data are contained within the article.

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
