# Peer review of "Phase-Controlled Tunable Unconventional Photon Blockade in a Single-Atom-Cavity System"

_micromachines, 2023, doi:10.3390/mi14112123_

Round 1
Reviewer 1 Report
Comments and Suggestions for Authors
Title: Phase-controlled tunable unconventional photon blockade in a single-atom-cavity system
Authors: Hong Li, Ming Liu, Feng Yang, Siqi Zhang, Shengping Ruan
In the submitted manuscript, Li et al. study theoretically unconventional photon blockade in a doubly pumped cavity-atom system. Their theory shows that using a combination of laser and squeezed light leads to a dependency of the photon blockade on the phase of the coherent field drive. This phase sensitivity is evident around the cavity-atom resonance and can be used to manipulate multi-photon contribution in the system emission by adjusting the phase and strength between the two excitation fields.
The authors use master equation modeling of their system in steady-state to study the unconventional photon blockade using an equal-time second-order auto-correlation function g^(2)(0). The simulation results can be well described with a simple analytical model, which is derived in the manuscript. The effect of the co-excitation of the cavity-atom system with laser and squeezed light has been recently studied in Phys. Rev. A 106, 023704 (2022) [reference 53 in the submitted manuscript], from where the analytical model was adjusted. Because of this continuity, a deeper understanding of the underlying mechanism of the destructive interference between excitation pathways originating from the parallel excitation with laser and squeezed light is expected. In this manuscript, the authors extend the study from Phys. Rev. A 106, 023704 (2022) to the system detunings from resonance and derive explicitly the phase dependency of the unconventional photon blockade. Despite the effect of the phase dependency is apparent from the presented data, the discussion of the destructive interference between excitation pathways, as the authors associate the effect with, is lacking.
The topic is certainly interesting and relevant for publication in the special issue of Chip Scale Quantum Technologies. However, the text clarity and readability, figure presentation, and results discussion have to be significantly improved first. After the improvements and addressing comments listed below, I will consider supporting the manuscript publication.
Clarity
Could authors comment on the choice of the used parameters of the cavity-atomic system? Ideally, could they refer to realistic experimental hardware having in mind?
Line 65: Authors use two laser detunings: Delta_a represents detuning in respect to resonance of cavity omega_c; Delta_c is detuning in respect to atomic resonance omega_a. I find the subscript definition very confusing - I would expect Delta_c=omega_c-omela_l, and similarly for Delta_a.
Comparing panels (a) in Figs. 2 and 3 for g/kappa=0.5 and Delta_c/kappa=0, the value should correspond to the same system settings, but g(2)(0) read from the figures is 1e-1 (Fig.2a) and 3e-1 (Fig, 3a). Please check if the values are consistent. Additionally, adding vertical lines to Figs. 2 and 3 corresponding to the datapoint g/kappa=0.5 and Delta_c/kappa=0 would tremendously help the reader to compare the figures.
Line 76: The meaning of superoperators in the Lindblad master equation is not discussed. Since these parameters have an essential effect on the cavity-atom interactions and thus directly affect the photon blockade effect, authors should introduce the symbols.
Eq 5: I would recommend using sign approximately, as done Phys. Rev. A 106, 023704 (2022). Additionally, I recommend explicitly stating that the first position in the ket notation |mn> corresponds to the atomic state, second to the photonic state.
Line 106: It is very unclear what the authors mean by “The interference of the two pathways leads to the single photon phenomenon.” Could authors rephrase the sentence? Also, could authors elaborate on the effect of “the interference of the two pathways”, which is essential for understanding the physics of this manuscript?
Lines 143-145: Please discuss more the comparison between the master equation simulations and your analytical model. Are these differences arising from different assumptions in the model description?
Could authors move Fig.1 closer to Equation 1, in my opinion, the understanding will improve significantly. I also recommend explicitly writing that authors assume the cavity-atom system pumped with double light sources: laser light and squeezed state generated with second-order nonlinear crystal. Right now, it is mentioned in the text very vaguely: “The system is driven by a complex pulsed laser, and the cavity exhibits second-order nonlinear crystals, the strength is U.”.
Other small mistakes
Line 32: I believe the word “interference” dropped out. Did the authors mean “The UPB mechanism arises from destructive interference between …”?
Line 50: Replace “second correlation function” with “second-order correlation function”. Please check that the “second-order correlation function” is used everywhere across the manuscript.
Line 57: The wording “The system is driven by a complex pulsed laser” is unclear. What do you mean by complex? Do you refer to complex amplitude?
Line 59: The meaning of Omega in eq. 1 is not explained.
Line 68: Photon cannot have a bunching effect. Bunching is a statistical property. Authors probably mean: “g2(0)>1 indicates the photon-bunching effect …”.
Line 70: Similar to above. Authors probably mean “g2(0)<1 indicates the photon antibunching effect”.
Line 88: Could authors check eq. 7? In my opinion, there is likely a missing phase term. It should be: left side = Omega exp (i varphi).
Fig1: The connection to the paper story is unclear. For better clarity, authors could highlight the two pathways discussed in lines 105 and 106 in Fig.1b. Also, authors could add labels to the ladder schemes in Fig. 1b to indicate which ladder is related to the cavity and which to the atom.
Fig2: Typo in the caption of the figure. The quantity on the x-axis is called normalized coupling strength.
Fig. 3: (i) There is an inconsistency between the caption and figures in the quantity on the x-axis. The caption uses “Delta_a/kappa”, the figure axes show Delta_c/kappa. (ii) Both panels of the figure are labeled as (a).
Fig. 4: (i) Both panels are labeled as (a). (ii) Could authors motivate why the color scale goes up to ~70 while their data in this figure reaches a maximum of ~10? If this is for easy color-scale comparison between Figs. 4 and 5, please indicate that in the text. Also, if possible, add to the figure (and to Fig. 5 as well) that the color scale represents g(2)(0).
Comments on the Quality of English LanguageThe text contains many typos (e.g., “duo to“ on line 116 instead of “due to”) and language errors. Below, I comment only on errors that could lead to reader confusion or misunderstanding.
Line 34: The phrase “nonlinear coupled cavity system” is ambiguine. Do the authors mean “system of a cavity coupled with a nonlinearity”? If so, I recommend writing: “The UPB effect has been predicted for the first time in the photonic molecular system [add reference], showing a nonlinearity-cavity coupled system.”
Line 35: The phrase “two weakly nonlinear coupled cavities” is unclear. Do you mean “two cavities which are coupled via a nonlinearity” or “two nonlinear cavities which are coupled”?
Line 101: I recommend replacing “produce” with “operates in”.
Reviewer 2 Report
Comments and Suggestions for Authors
The authors study the phenomenon of unconventional photon blockade in a system composed of a nonlinear oscillator coupled to a two-level system, when driven by a weak complex field. They particularly investigate the effect of the field phase on the second order correlation function quantifying photon antibunching. They show that in the presence of nonlinearity this phase can considerably affect the correlation function. The article possesses merits which warrant publication, but before we recommend it, we ask the authors to consider the following remarks:
1. The authors need to improve the language of the manuscript and correct the many typos. For example, pg. 1 line 32 destructive interference, \chi^{(2)} for the nonlinearity etc. Also, they use some unusual wording for various quantities, see for example pg. 2 lines 63-64. Please rewrite the article using proper language and terminology, otherwise I don’t see how it can be published here or elsewhere.
2. In the correlation function in Eq. (2) there is a square missing in the denominator. Also, in Fig. 3, the horizontal axis label should be \Delta_a/\kappa.
3. In the discussion of Fig. 2(b) the authors should emphasize that in the presence of nonlinearity the choice of the field phase considerably affects the correlation function.
4. In Fig. 3(b) it appears that the correlation function is closer to zero for \phi=1.2 rad, why do the authors say that strong antibunching is observed for \phi=1.8 rad? It’s probably copy-paste from the comment regarding the previous figure.
5. When the authors say numerical solution of the master equation what exactly do they mean? That they use a wavefunction with more states in (5)?
6. The authors may would like to cite the following articles studying dynamical photon blockade:
(a) S. Ghosh and T. C. H. Liew, Dynamical blockade in a single-mode bosonic system, Phys. Rev. Lett. 123, 013602 (2019).
(b) D. Stefanatos and E. Paspalakis, Dynamical blockade in a bosonic Josephson junction using optimal coupling, Phys. Rev. A 102, 013716 (2020).
(c) M. Li, Y.-L. Zhang, S.-H. Wu, C.-H. Dong, X.-B. Zou, G.-C. Guo, and C.-L. Zou, Single-mode photon blockade enhanced by bi-tone drive, Phys. Rev. Lett. 129, 043601 (2022)
Comments on the Quality of English LanguageSee the review
Round 2
Reviewer 1 Report
Comments and Suggestions for Authors
Title: Phase-controlled tunable unconventional photon blockade in a single-atom-cavity system
Authors: Hong Li, Ming Liu, Feng Yang, Siqi Zhang, Shengping Ruan
The authors in the re-submitted manuscript address my comments and implement my suggestions. These small changes in the text significantly improved the text and figures' readability. I have two more comments connected to the changes made to the manuscript.
Comment 1: I want to bring the attention of the authors to the color scale of Figures 4 and 5. Could they indicate either in the figure captions, better as part of the color scale, that the color scale is logarithmic? Now, it is not mentioned and could lead to confusion.
Comment 2: Authors in the reply specify their suspicion for the origin of the difference between their analytical model and quantum master simulations. They mention numeric precision and different model assumptions. In my opinion, the reader could benefit from such information being added to the text. Below, I include the answer I am referring to.
From review (round 1):
Question (reviewer 1, review round 1):
Please discuss more the comparison between the master equation simulations and your analytical model. Are these differences arising from different assumptions in the model description?
Authors answer:
We had a proper discussion. I think some of the differences may be due to program accuracy and different assumptions in the model description.
If the authors add the above, I support the publication.
--------
Below, I list some statements I noticed through reading where phrasing could be improved. The decision about the implementation is up to the authors.
Line 196: Authors added to conclusions a sentence: "By selecting different parameters, the optimized parameters can be found to realize the photon blockade effect." I discourage the authors from using "different parameters" phrasing, as it raises the question that there are better parameters than the authors discuss in the paper. Authors probably mean: "The photon blockade effect can be improved by close assessing of the system parameters, as shown in this paper."
Lines 179-179: I still struggle with formulation "Therefore, the numerical results obtained by solving the master equation are almost in agreement with the analytic results of the second-order correlation function." Quantum master equation is, in my opinion, assumed as a standardized tool for tasks discussed in the paper. Therefore, the results from this method could be seen as "standard" to which the simplified analytical model should be compared, not the other way around. Could the authors rephrase this sentence to reflect on comparing the analytical model to the quantum master equation?
End of line 139: typo, the authors mean "the quantum master equation"
Author Response
Please see the attachment。
